# Scaffold-Lab: Critical evaluation and ranking of protein backbone generation methods in a unified framework

**Zhuoqi Zheng**◉°, **Bo Zhang**°, **Bozitao Zhong, Jinyu Yu, Kexin Liu, Zhengxin Li, Junjie Zhu, Ting Wei**\*, **Hai-Feng Chen**◉\*

State Key Laboratory of Microbial metabolism, Joint International Research Laboratory of Metabolic Developmental Sciences, Department of Bioinformatics and Biostatistics, National Experimental Teaching Center for Life Sciences and Biotechnology, School of Life Sciences and Biotechnology, Shanghai Jiao Tong University, Shanghai, China

☯ These authors contributed equally to this work.
\* weitinging@sjtu.edu.cn (TW); haifengchen@sjtu.edu.cn (H-FC)

## Abstract

Recent advances in *de novo* protein design have significantly progressed, particularly in protein backbone generation, which remains a challenging yet valuable task. However, there has been a lack in standardized evaluation framework to accurately assess diverse methods, nor is there a comprehensive analysis of their practical applications. In this study, we proposed Scaffold-Lab, a unified framework for systematic evaluation of protein backbone generation methods, encompassing a wide range of metrics including designability, novelty, diversity, efficiency and structural properties. We thoroughly evaluated seven representative methods, providing a detailed analysis of their performance and utilities. Our findings highlight that generating long proteins for unconditional generation and accurately reconstructing motifs are key bottlenecks for most methods. These results underscore both the strengths and limitations of existing protein backbone generation methods, offering new insights for further development and improvement in this field.

## Author summary

Designing novel protein structures holds significant potential in protein engineering and drug design. With the emergence of deep learning based generative methods applied to protein structure generation, it is crucial to evaluate these methods efficiently in a comparative manner, benefiting both method development and the downstream applications of protein design. In response, we propose Scaffold-Lab as a unified evaluation framework featuring a suite of metrics that span the quality and structural properties of designed proteins. We evaluated five and four methods in terms of unconditional generation and motif-scaffolding both qualitatively and quantitatively. Our findings show that

**Data availability statement:** The code for Scaffold-Lab evaluation framework is available at https://github.com/Immortals-33/Scaffold-Lab. A Colab Notebook for easy-to-use evaluation is provided at https://colab.research.google.com/github/Immortals-33/Scaffold-Lab/blob/main/scaffold_lab.ipynb. The detailed guidelines used to reproduce the experiments in our manuscript is available at https://github.com/Immortals-33/Scaffold-Lab/tree/main/baselines. The original backbones generated could be accessed via https://zenodo.org/records/20080699.

**Funding:** National Key Research and Development Program of China (2025YFA0921000 and 2023YFF1205102), the Fundamental Research Funds for the Central Universities (YG2023LC03), and the National Natural Science Foundation of China (32571435).The funders had no role in study design, data collection and analysis, decision to publish, or preparation of the manuscript.

current methods performing generally well in unconditional generation with bias towards certain types of local structure such as alpha helices. The analysis of motif-scaffolding revealed that the reconstruction of motifs is a more difficult task than generating the designable scaffolds. In conclusion, we highlight the strengths and limitations of current methods and offer insights for future developments and applications with protein backbone generation.

## Introduction

Proteins are fundamental to biological systems due to their diverse functions, largely attributed to the hierarchical organization progressing from amino acid sequences to three-dimensional (3D) structures [1,2]. Although substantial kinds of proteins already exist in nature, they represent a limited number of the whole protein space [3]. Computational protein design has been developed to expand the functional repertoire of proteins. The magnificent breakthroughs in protein design and protein structure prediction [4] have allowed biologists to explore protein structures in greater depth, breathing new life into the fields of protein science and engineering [5,6].

Protein design has undergone a paradigm shift over the past decades. Early *de novo* protein design methods relied on physics-based methods to search through rotamer libraries in order to design new protein sequences based on a pre-defined structural scaffold [7]. In recent years, the rapid development of deep learning has significantly advanced computational structural biology, especially in protein sequence design [8–15] and protein structure prediction [4,16–21]. Given that proteins typically realize their biological functions directly by 3D structures [1,22], protein backbone generation has been carried out as a new approach to directly manipulate the topologies of proteins [23,24]. While accurately modeling the structural constraints in 3D Cartesian space is challenging for physics-based methods, the advancements in generative models such as denoising diffusion models [25,26]. have revolutionized the field with their exceptional performance in modeling high-dimensional data distribution [27], which has already achieved incremental progress [28–34].

Although several methods have tested the designed protein backbones in wet lab [34,35], experimental validation lags far behind computational generation in throughput. An *in silico* evaluation pipeline could accelerate the selection of possible candidates to be committed with wet-lab resources. Furthermore, due to the varying evaluation metrics employed by different methods, there is an urgent need for standardized benchmarks to assess protein backbone generation methods. This need is especially pressing when compared to the various established benchmarks in place for sequence design [36–39], protein fitness prediction [40] and molecular generation [41]. While most methods report their performance against previous ones, differences in the definition and thresholds of evaluation limit effective comparisons. Additionally, there is no standardized evaluation suite for assessing methods from scratch. Since protein backbone generation is a crucial step in *de novo* protein

design and protein engineering, a better understanding of how these methods can be applied in real-world contexts remains a key gap in the field.

To address these challenges, we proposed Scaffold-Lab, a unified framework for the comprehensive evaluation of protein backbone generation methods. The framework is divided into two primary tasks. *Unconditional generation* involves creating protein scaffolds with user-specified lengths with no other restrictions. This task serves as proof-of-concept for protein backbone generation methods, demonstrating how well these models learn the underlying distribution of realistic protein structures. *Conditional generation*, on the other hand, generates proteins under specific constraints defined by users. This task is vital as it represents the goal of a fully-programmed protein design roadmap, aiming to create de novo proteins with properties determined by researchers. We selected *motif-scaffolding* task as a representative example, where the objective is to generate novel scaffolds around one or multiple motifs extracted from native protein structures, which usually plays a key role in specific biological functions. This strategy is particularly useful in applications like designing novel enzymes [42], binders [34,43,44] or vaccines [45] and has been widely explored and evaluated within the last few years [33,34,46–49].

Seven representative protein backbone generation methods were evaluated based on their performance and availability. They were grouped into categories based on the lengths of generated proteins. We recast various evaluation metric into this evaluation framework and ranked different methods to assess their capabilities.

Beyond prevalent computational metrics, we introduced a personalized evaluation scheme to gain a deeper understanding of the progress and limitations of these methods. Structural properties analysis for unconditional generation revealed that current methods tend to produce highly structured proteins. For motif-scaffolding, we went beyond traditional metrics to investigate the underlying reasons behind the different outcomes of designs. We observed the task of motif reconstruction is a more difficult task than the task of generating a designable scaffold, and that creating a successful overall scaffold is a prerequisite for effective motif handling. In summary, our research undertook a systematic assessment using a novel framework to elucidate the current progress and limitations in this field, paving the way for future developments.

## Material and methods

Table 1 lists the methods and their corresponding tasks selected for evaluation in the Scaffold-Lab benchmark test. Both *Genie* [50] and *FrameDiff* [53] incorporated the invariant point attention (IPA) module introduced by Jumper et al. [4] as a key component of their architectures. *FrameDiff* utilized diffusion modeling in Riemannian manifolds to perform protein diffusion in SE(3) space, while *RFdiffusion* [34] integrated diffusion models with a pretrained structure prediction model RoseTTAFold [19] to greatly improve the performance. *Chroma* combined diffusion models with graph neural networks

**Table 1. Methods selected to be tested in Scaffold-Lab.**

| Method | Released Date | Category† | Software Link | Tasks evaluated |
|---|---|---|---|---|
| Genie [50] | 2023-01 | Diffusion-based | https://github.com/aqlaboratory/genie | Unconditional |
| FrameDiff [50] | 2023-03 | Diffusion-based | https://github.com/jasonkyuyim/se3_diffusion | Unconditional |
| RFdiffusion [34] | 2023-04 | Diffusion-based | https://github.com/RosettaCommons/RFdiffusion | Both |
| TDS [47] | 2023-06 | Diffusion-based | https://github.com/blt2114/twisted_diffusion_sampler | Conditional |
| GPDL [49] | 2023-10 | ESM-based | https://github.com/sirius777coder/GPDL | Conditional |
| FrameFlow [51] | 2023-10 | Flow-based | https://github.com/microsoft/protein-frame-flow | Both |
| Chroma [35] | 2023-11 | Diffusion-based | https://github.com/generatebio/chroma | Both |

†We followed the category introduced by Zhang et al. [52].

‡M = million; B = billion.

(GNNs) to simultaneously generate protein structure and sequence. *TDS* [47] used a particle filtering strategy to raise conditional sampling applied to motif-scaffolding task. *FrameFlow* replaced the denoising score matching processes in diffusion models with a flow-matching process to significantly elevate the efficiency and designability compared with previous works. *GPDL* [49] combined ESMFold [16] with constrained hallucination and inpainting strategies to generate highly designable and diverse scaffolds around given motifs. Table 2 outlines the grouping strategy of unconditional generation by lengths. All analysis in this benchmark were performed under this framework. Further details of dataset and the grouping schemes are provided in Supplementary Materials.

The overall workflow of Scaffold-Lab is shown in Fig 1. Each of the generated backbone was evaluated with a refolding pipeline. We generated 100 backbones with 10 sequences for each backbone to be evaluated. ProteinMPNN [12] was used as the sequence designer throughout the pipeline, as it has been widely utilized and validated by wet-lab experiments [34,46,54]. A comparison of ProteinMPNN to another state-of-the-art method, ESM-IF1 [55] in S1 Fig in S1 File indicated that ProteinMPNN outperformed ESM-IF1 in terms of refolding ability, thereby maximizing the potential of different protein backbone generation methods. For structure prediction, both ESMFold [16] and AlphaFold2 have been shown to efficiently distinguish designable backbones [56]. We chose ESMFold as the standard refolding predictor due to its speed

**Table 2. Length grouping of unconditional generation.**

| Group | Length | Method |
|---|---|---|
| Short | 50, 100 | Genie, FrameFlow, FrameDiff, Chroma, RFdiffusion |
| Medium | 200, 300, 400, 500 | FrameDiff, Chroma, RFdiffusion |
| Long | 600, 800, 1000 | Chroma, RFdiffusion |

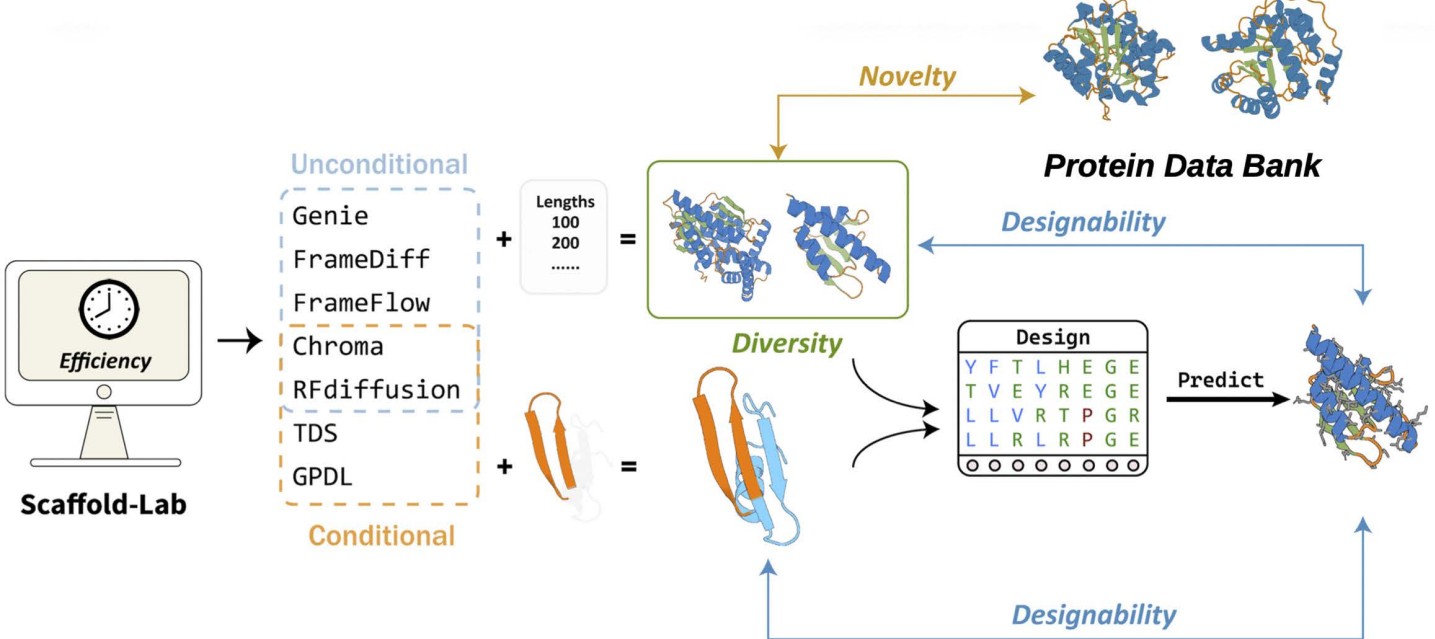

**Fig 1. The overall workflow of Scaffold-Lab.** Seven state-of-the-art protein backbone generation methods were evaluated by two types of tasks throughout a refolding pipeline. Four qualitative metrics were introduced to qualitatively assess the performances of methods, namely designability, novelty, diversity and efficiency.

and competitive performance on long proteins [57]. Additionally, a comparison of a subset of sequences between Alpha-Fold2 [4], ESMFold [16] and OmegaFold [58] revealed a relatively strong correlation among the three structure prediction methods. (S2 Fig in S1 File).

For unconditional generation, we generated 100 backbones at each unique evaluated length for each method. For motif-scaffolding, we generated 100 backbones for each of the 24 benchmark cases of different methods. Ten sequences were then designed with ProteinMPNN for each backbone and then predicted with ESMFold to calculate different metrics of each backbone.

## Evaluation metrics

We divided evaluation metrics into five categories, namely *designability, novelty, diversity, efficiency* and *structural properties*. The first four categories are quantitative that were used for ranking different methods. While most of the these are *in silico* metrics, they serve as valuable proxies for high-throughput *in vitro* experiments and provide valuable insights for selecting candidates in downstream applications of protein design.

## Designability

*Designability* is the most critical metric for protein backbone generation tasks as it assesses whether there exists a sequence that can fold into the generated protein backbone. We employed the self-consistency TM-score (*sc-TM*) [33] where a protein backbone is considered as designable if the refolded structures and the original backbones shares a TM-score [59] > 0.5, indicating that they have the same fold pattern [60]. Many prior studies also used the success definition of self-consistency RMSD (*sc-RMSD*) < 2.0 Å instead of sc-TM > 0.5 [34,61].

The success criterion of sc-RMSD may be more stringent than that of sc-TM. While this is sensible for relatively small proteins that can be handled by most existing methods, RMSD performs poor as the size of the protein increases. In this case, the original success definition of sc-RMSD < 2.0 Å becomes less discriminative (S3 Fig in S1 File). To address this, we implemented the following well-curated evaluation strategies:

1. We use the success definition of sc-TM > 0.5 for unconditional generation (protein length ranges from 50 to 1000). This provides a clear comparison of the performance of different methods across varying protein lengths, especially for longer proteins. For evaluation on motif-scaffolding, we use sc-RMSD < 2.0 Å since designed proteins here are typically under 150 residues, providing a more rigorous test of method performance.

2. To complement the stringency of existing metrics, we develop a "Top $N$ strategy". While current methods typically consider the best sequence for evaluating designability, we propose that the number of sequences that fold into a generated structures also reflects the robustness and fitness of backbones [62–64]. A protein structure that accommodates multiple sequences is more stable against mutations and thermodynamically more stable [65], which are important factors in experimental selection. Thus, we calculated the mean sc-TM (or sc-RMSD) of the top $N$ sequences (with highest sc-TM or lowest sc-RMSD), defined as:

$$Success^{TM-score} = \begin{cases} 1, \frac{1}{N}\sum_{i=1}^{N} scTM > 0.5 \mid (scTM_1 > \ldots > scTM_N > scTM_i) \\ 0, \, others \end{cases} \tag{1}$$

$$Success^{RMSD} = \begin{cases} 1, \frac{1}{N}\sum_{i=1}^{N} scRMSD < 2.0 \text{ Å} \mid (scRMSD_1 < \ldots < scRMSD_2 < scRMSD_i) \\ 0, \, others \end{cases} \tag{2}$$

Here we chose $N = 3$ for evaluation and reported the distribution of sc-TM and the success rate by picking the best sequence as a fair comparison (shown in Figs 2B and S5- S6 in S1 File).

3. We applied dynamic thresholds of sc-TM and sc-RMSD for success definitions and evaluated the trends in success rates across different methods. This approach provides a novel prospective on method performance without relying on subjective thresholds, further enhancing the comprehensiveness and reliability of evaluation.

## Novelty

*Novelty* measures the structural difference between the generated protein backbones to all existing proteins. We followed the approach in FrameDiff [53] by calculating the highest TM-score between generated protein backbones and all proteins in Protein Data Bank (PDB) [67,68], which is referred to as *pdb-TM*. We used Foldseek [69] to search for the nearest protein structure in the PDB database that matched the generated backbones.

We also observed that some generated backbones, although novel, were undesignable. To address this, we introduced a penalty for poorly designable backbones when calculating the *novelty score*, which is formulated as below:

$$S_{novelty} = \begin{cases} pdbTM + \left(0.5 - w\frac{1}{n}\sum_{i=1}^{n} scTM\right), \frac{1}{n}\sum_{i=1}^{n} scTM < 0.5 \\ pdbTM, \frac{1}{n}\sum_{i=1}^{n} scTM > 0.5 \end{cases} \tag{3}$$

Where $n$ refers to the "Top N" strategy when selecting sequences generated by ProteinMPNN, and $w$ means the degree of penalty. A higher $w$ imposes a more stringent penalty on backbones with less designability. We set $n = 3$ and $w = 1$ throughout our experiments. If the average sc-TM value for a sequence is greater than 0.5 (indicating it is designable), we retain the original pdb-TM value. Otherwise, a penalty is applied based on the shortfall between sc-TM and the 0.5 threshold.

## Diversity

*Diversity* measures the internal variation among generated samples, reflecting the determinacy and the generalizability of models. In the context of protein backbone generation, a method that exhibits greater diversity indicates that it can generate a broader range of structural topologies. This heterogeneity provides more options for selecting candidates for the subsequent sequence design stage, which can benefit both potential applications and the success rate of wet-lab experiments.

Since diversity is a referent metric of internal differences within the generated examples, clustering the generated protein backbones is a natural approach for assessing the similarities between them. We grouped the protein backbones by length and used Foldseek-Cluster [70] to cluster each group of protein backbones. We used a clustering TM-Score threshold of 0.5 based on the aforementioned consensus [60]. For the diversity definition, we followed the approach from Yim *et al.* [53] by calculating the number of unique clusters divided by the total number of protein backbones each group:

$$Diversity = \frac{N_{unique\ clusters}}{N_{total\ number\ of\ proteins}} \tag{4}$$

In general, a higher count of clusters reflects more heterogeneous topologies appeared within a subset of sampled protein backbones.

## Efficiency

Compared with protein sequence design, the modeling of three-dimensional structures often results in a higher time and computational resource consumption. Given that efficiency is critical for high-throughput protein engineering, we compared the runtime of different protein backbone generation methods. For this comparison, we generated 100 backbones

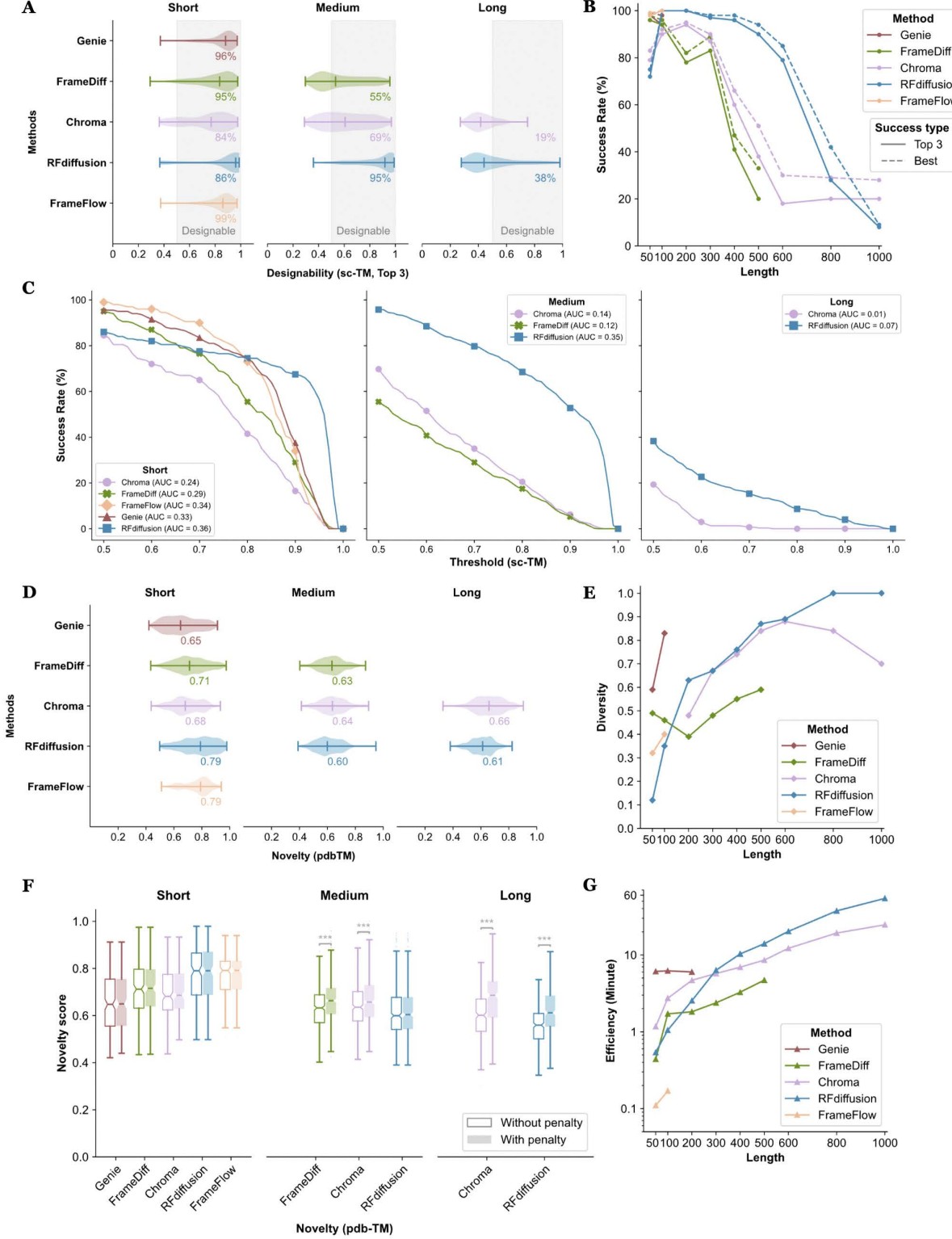

**Fig 2. Evaluation results of unconditional generation. 100 backbones were generated for each unique length for different methods. (A)** Distribution of top 3 sc-TM. The number attached to each group denotes the proportion of backbones with top 3 sc-TM ≥ 0.5. **(B)** Success rates of tested

methods. Success rate is defined by the number of backbones passing the sc-TM threshold divided by the total number of backbones within each group, with top 3 sc-TM by solid line and best sc-TM by dashed line. The text of numbers refers to the solid lines. **(C)** AUC of dynamic sc-TM thresholds. The values on x-axis denotes the designability threshold of different sc-TMs, where the success definition is "top 3" sc-TM> $k$ when x = $k$. We take the step of sc-TM as 0.01 during the calculation and used *Scipy* [66] to calculate the area under the curve. The AUC values of different methods are shown inside the legend of each subplot. **(D)** Distribution of pdb-TM values. The number attached to each group denotes the median value. **(E)** Diversity performances. Diversity is defined as the number of unique clusters divided by the total number of proteins within each group using Foldseek-Cluster. **(F)** Differences between novelty calculation by whether adding the penalty term of designability. The colors of different boxplots are indicators of different methods consistent with other plots in Fig 2. Stars on each data group indicate the differences between two types of data within the group, calculated using the T-test. *, $p$-value<0.05; **, $p$-value<0.01; ***, $p$-value<0.001. **(G)** Time usage for generating protein backbones among different methods. The efficiency is calculated as the average time usage per backbone for generating 100 backbones. All tests were conducted under identical hardware resources.

per group and calculated the average time taken to generate each backbone. All experiments were conducted under the same hardware resources to maintain consistent comparisons.

## Structural properties

Proteins exhibit various forms of internal structures that are organized in a hierarchical way. A protein in its desired folded state usually has unique covalent and non-covalent interactions, which are crucial for stabilizing the protein and driving it towards its energy-favored state [2]. Since protein backbone generation mainly focus on three-dimensional structures, it is essential to assess the physical plausibility and quality from structural perspectives. Thus, we have analyzed the structural properties of protein from two different aspects, *secondary structure elements*, *radius of gyration* and *structural validity.*

The secondary structure element (SSE) is a key feature of proteins, directly representing the type of topologies and structural organization of a protein backbone. We also computed the radius of gyration ($Rg$) for each generated protein backbones. $Rg$ is a critical property of biological polymer molecule, as it measures the size or compactness of proteins. $Rg$ can also be used to distinguish between structured proteins and IDPs, which is of great significance to the protein functions [71]. Finally, structural validity are used to examine the biophysical properties of generated proteins using Mol-Probity [72].

## Ranking

For the ranking of different methods, we used the widely recognized Technique for Order Preference by Similarity to Ideal Solution (TOPSIS). TOPSIS [73] is a multi-criteria decision analysis method based on the principle that the best alternative is the one that has the shortest geometric distance to the positive ideal solution, and the farthest from the negative ideal solution. This method is commonly combined with comprehensive factor weighting methods to improve the reliability of evaluation. These weighting methods can be subjective, such as the Analytic Hierarchy Process (AHP) [74], or objective, like the CRiteria Importance Through Intercriteria Correlation (CRITIC) [75], which has been successfully applied in the evaluation of protein sequence design methods [38]. We have included a brief mathematical description of the ranking methods in Supplementary Materials.

## Results

### Unconditional generation

**Designability.** The overall results of unconditional generation task are presented in Fig 2. We first computed the "top 3" sc-TM distribution across various protein length groups (Fig 2A). Nearly all evaluated methods performed well for short proteins, but performance declined as the lengths gradually increased. Eventually, neither the median sc-TM value of *RFdiffusion* nor *Chroma* reached the threshold of 0.5 for longer proteins. It is worthwhile to note that *FrameFlow*, *Genie*

and *FrameDiff* generated short proteins that were almost entirely designable, while *RFdiffusion* demonstrated the most consistent performance across all lengths.

The corresponding success rates are shown in Fig 2B. where it is clear that the success rate generally decreases as the protein length grows. A notable turning point was observed at protein lengths of around 300 residues. For proteins with 300 or less residues, all the tested methods performed strongly, with success rates around or exceeding 80%. However, as protein length increased beyond 300 residues, success rates dropped rapidly, falling below 30% for proteins with 800 or more residues. Additionally, the comparison between the two success rate definitions indicates that the robustness of the designed backbones also decreases to some extent as protein length increases.

Although sc-TM is an effective indicator for the quality of generated backbones, using a static threshold may introduce subjectivity, especially in downstream applications, such as the selection of generated backbones for wet-lab experiments. To address this issue, we introduced dynamic choice for threshold choices for TM-score in defining success and added the Area Under the Curve (AUC) as an additional measure of designability, with results displayed in **Fig 2C**. A method with a higher AUC value indicates better performance, regardless of the subjective definition of designability. In this context, we observed that *RFdiffusion* outperformed all other methods across all categories. This is primarily due to its reliable performance when the success definition becomes more stringent, as it maintains success rates of around 70% and 50% for short and medium-length proteins, respectively. In comparison, other methods showed steady decline as the sc-TM threshold was raised.

Overall, *RFdiffusion* performed the best among all tested methods, while other methods also demonstrated reliable performance on short proteins. We argue that the outstanding performance of *RFdiffusion* largely benefits from its integration with the pre-trained structure prediction model RoseTTAFold, which provides a robust approach to capturing the underlying physical patterns of native protein structures, resulting in more reasonable protein backbones. This hypothesis is supported by its direct comparison with *FrameDiff*, where both methods use a similar SE(3) diffusion process and frame parametrization, but exhibit distinct performances in terms of designability. This highlights the significant role of pre-trained structure prediction models in protein backbone generation. On the other hand, lightweight models without extensive pretraining networks offer a balance between efficiency and designability, while also encouraging exploration of advanced techniques for modeling protein structures.

Our results on both static and dynamic designability threshold also provides a flexible framework for evaluating different methods across various scenarios, including method development and the downstream selection of proteins for wet-lab experiments. The sc-RMSD values further support this conclusion (S4 Fig in S1 File). As a reference, we also calculated the best sc-TM and sc-RMSD distribution, shown in S3–S6 Figs in S1 File. Additionally, the AUC values for sc-TM (selecting best sequence) and sc-RMSD (both using best sequence or "top 3" sequences) are provided in S7–S9 Figs in S1 File.

**Novelty.** As shown in **Fig 2D**, the pdb-TM values predominantly range between 0.6 and 0.7, with no significant trends observed across the different methods or length categories. Notably, *FrameFlow* and *RFdiffusion* exhibit slightly highest pdb-TM values in the group of short proteins, suggesting a tendency to generate protein backbones more similar to natural proteins. Given their relatively higher success rates in corresponding categories shown in Fig 2B, we hypothesize that this trend in novelty may also reflect the fitness of model, indicating a potential trade-off between novelty and designability.

To further validate this hypothesis, we used the novelty score, where penalty were applied to protein backbones with low designability, to explore the correlation between these two factors. **Fig 2F** shows the differences in the distribution of pdb-TM and novelty score, where we conducted a one-sided Mann-Whitney U test [76] across different groups, distinguished by whether or not the penalty term was applied. The results indicate significant differences in some groups with *p-values* < 0.001, including *FrameDiff*, *Chroma* in medium-protein group ($6.11 \times 10^{-8}$ *and* $5.19 \times 10^{-4}$, respectively) and *RFdiffusion, Chroma* in long-protein group ($2.46 \times 10^{-15}$ *and* $3.33 \times 10^{-16}$, respectively).

Interestingly, *RFdiffusion* and *FrameFlow* in the short protein group did not show significant differences in results with or without penalty, implying most proteins within those groups pass the designability threshold, which aligns with our

previously proposed hypothesis. Furthermore, the differences between the penalty and non-penalty groups increased as protein length grew, indicating the appearance of protein backbones with high novelty but low designability. Although different methods did not exhibit strong variations in novelty overall, our analysis suggests that designability should be partially considered when calculating novelty. The complete results of significant test are provided in S2 Table in S1 File.

**Diversity.** The results in Fig 2E indicate that *FrameFlow* had the lowest diversity compared with other methods, which probably owes to the diversity-designability trade-off mentioned in Yim *et al*. [53]. *RFdiffusion* and *Genie* both showed a continuous increase in diversity by length as protein length increased within their respective length ranges. *RFdiffusion*, started from a lower point, reached near-maximal diversity when the protein length exceeded 800 residues. Interestingly, *FrameDiff* and *Chroma* displayed non-monotonic behavior: *FrameDiff* initially declined before increasing, while *Chroma* exhibited the opposite trend. Overall, diversity tended to increase with protein length, potentially due to the distinctive levels of redundancy in the training set at different lengths.

**Efficiency.** The result displayed in **Fig 2G** show significant differences in efficiency among various methods, where the greatest variance observed when generating short proteins. We also noted that the growth trend of *Chroma* stabilized when generating longer proteins, benefiting from its scaling technique [35], which is particularly useful for generating large proteins. However, it is important to note that the processing speeds of the current methods are still in the range of several minutes, which remains a gap compared to protein sequence design methods that can generate sequences in just a few seconds. Improving inference speed and modeling efficiency will be an important direction for future development in this field.

## Analysis of structural properties

**Secondary structure analysis.** **Fig 3A** illustrates the composition of secondary structure elements (SSEs) based on average frequency. The results show that most methods tend to generate structured proteins, with a $\alpha$-helices comprising more than 50% of the structures in most cases. We also noticed that this preference for structured proteins remain consistent even as protein length increases up to 1000 residues. This is probably due to the imbalanced training set where loop-abundant structures only account for a small subset of training data, which could be a key factor driving this behavior.

The density distribution of SSEs for individual proteins shown in **Fig 3B** revealed that the distribution of short protein group is broader than that of longer groups. *RFdiffusion* demonstrated a strong tendency to generate proteins primarily composed of $\alpha$-helices. However, as the protein length increases, *RFdiffusion* also produced proteins with more diverse SSEs than *Chroma*. This distribution, combined with the overall trend, suggests that the methods generally favor generate helical proteins. This preference is likely due to the abundant presence of $\alpha$-helices and the fact that short-range interactions (which are common in $\alpha$-helices) are easier for models to learn, in contrast to the long-range interactions found in β-sheets and more irregular regions like loop contents.

**Radius of gyration.** The radius of gyration (Rg) was also calculated due to its importance in revealing the compactness of proteins. Results are shown in **Fig 3C**. Using empirical regression curve of Rg for structured proteins as a reference, we observed that most methods tend to generate proteins with an Rg value below that of the corresponding length range, indicating the compactness of generated proteins. *Chroma* was the only exception as it generates more stretched backbones, particularly for longer proteins, suggesting a potential to design more flexible protein structures.

Overall, most of the tested methods demonstrated a significant tendency to generate highly structured regions. We hypothesize that the training set which excludes most loop-abundant structures, may be a key factor to this behavior. While focusing on highly structured regions allows backbone generation methods to better capture the distribution of protein topologies, this approach may limit the modeling of more irregular regions like ordered loops and intrinsically ordered regions, both essential for regulating protein functions and enzyme catalytic activities [79]. Diffusion models have lately been applied to model intrinsically disordered regions in both protein design [48] and protein ensemble generation [80].

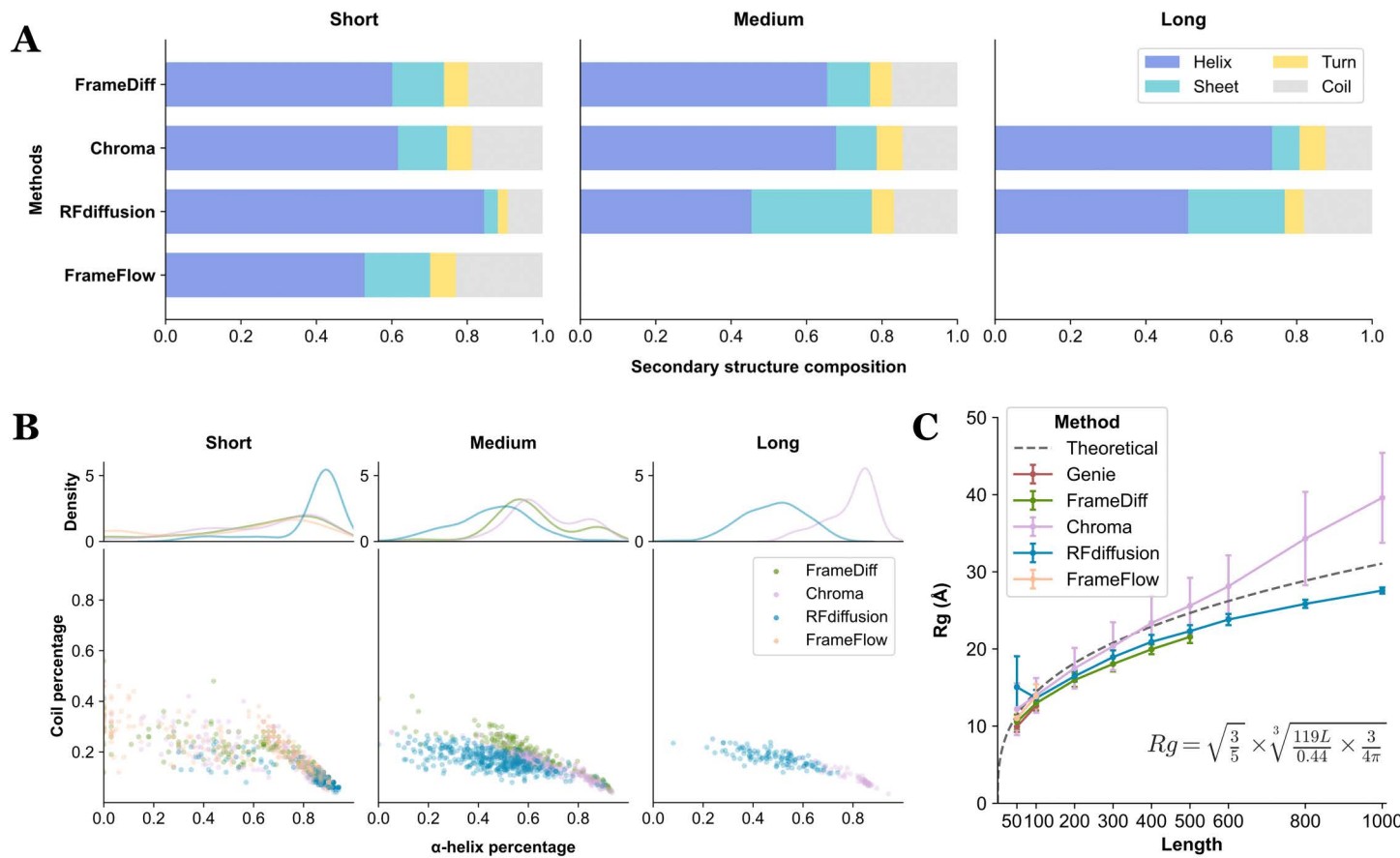

**Fig 3. Structural analysis on generated proteins. (A)** Secondary structure distribution averaged by all backbones of the group within each method. **(B)** Distribution of secondary structure composition of each protein. The analysis was performed onto the original backbone generated from each method. The SSEs per residue are based on the assignment from DSSP, where α-helices are defined as the residues annotated with "H", "G" or "I"; "coil are defined as the residues annotated as "C" or "S". *Genie* was excluded from this test as it solely generates $C\alpha$-only backbones. **(C)** Radius of gyration (Rg) values of different methods. The points are the mean value of Rg and the error bars denote the standard deviation interval within each group. The empirical regression curve of Rg for structured proteins is shown in a dashed style with the text denoted the formula based on previous studies [77,78].

We believe that improving modeling techniques for loop regions could alleviate this limitation and enhance the utility of these methods in real-world protein design applications.

**Structural validity.** While most of the methods are capable of generating protein backbones with idealized secondary structures and idealized globular shape, the local geometric details of inter-residue contacts and rationality of biophysical properties are yet to be explored. Thus, we examined the structural validities of tested methods with MolProbity [72] in terms of clash score and the percentage of bad angles and lengths. It could be observed from S10 Fig in S1 File that the clash scores of different methods increased over the length of generated backbones and the overall distribution is higher than the idealized values of high-quality referenced crystal structures [72], with the exception of RFdiffusion obtaining a significantly lower clash score than different methods. However, in terms of bad bonds and bad angles, Chroma performed the best among all tested methods with the distribution close to zero, with the others having a percentage ranging from 0% to 20%, far beyond the expected threshold of 0% for bad bonds and 0.1% for bad angles, respectively. (S11–S12 Figs in S1 File).

We hypothesize the differences between methods in terms of various aspects of structural validity may be attributed to the differing training schemes of the methods tested, as well as the different representation schemes of generated backbones. For instance, Chroma has an internal sequence design module which is capable of generating backbones with sidechain atoms, which might contribute to its high clash scores. Additionally, this brings out the fact that beyond the idealized overall protein structures, the local biophysical properties of current generative models still have room for improvements. This could be achieved by introducing auxiliary losses specialized for optimizing geometric details locally for both inter-residue and intra-residue contacts.

### Ranking of different methods

After analyzing the generated protein backbones, we employed a ranking procedure based on a procedure based on the TOPSIS decision-making method combined with the CRITIC weighting-selection technique. The results are summarized in Fig 4 and Table 3. We ranked the methods according to three length categories: short, medium and long proteins.

We found *FrameFlow* and *Genie* performing the best on short proteins and *RFdiffusion* showing outstanding performance for long proteins. We also noted that no single method consistently performed the best or worst across all groups, indicting that each method excels in different scenarios. Therefore, we recommend a flexible and personalized selection of methods in practical applications, tailored to achieve the most optimal performance for specific circumstances.

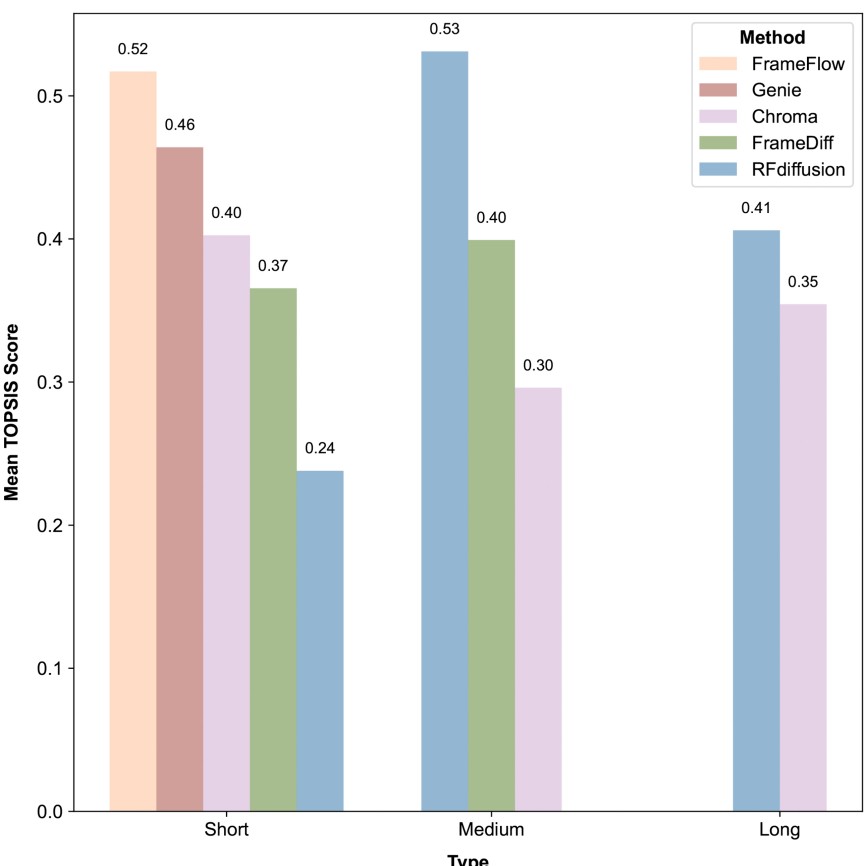

**Fig 4. Mean TOPSIS score of different methods.** The numbers upon each bar denotes the mean value across different lengths within the length group.

Table 3. **Ranking of Different Methods.** Four metrics with arrows are shown as mean values within each group without data preprocessing. $C_i$ denotes the calculated relative nearness value of TOPSIS results, where higher indicates better performances.

| Type | Method | Length | Designability(↑) | Novelty(↓) | Diversity (↑) | Efficiency (↓) | $C_i$ | Mean $C_i$ | Rank |
|---|---|---|---|---|---|---|---|---|---|
| Short | Chroma | 50 | 0.691 | 0.715 | 0.54 | 1.17 | 0.368 | **0.403** | 3 |
| | | 100 | 0.763 | 0.682 | 0.66 | 2.73 | 0.437 | | |
| | FrameDiff | 50 | 0.781 | 0.731 | 0.49 | 0.44 | 0.378 | **0.365** | 4 |
| | | 100 | 0.799 | 0.711 | 0.46 | 1.71 | 0.353 | | |
| | FrameFlow | 50 | 0.807 | 0.812 | 0.32 | 0.11 | 0.509 | **0.517** | 1 |
| | | 100 | 0.862 | 0.721 | 0.4 | 0.17 | 0.525 | | |
| | Genie | 50 | 0.841 | 0.715 | 0.59 | 7.92 | 0.377 | **0.464** | 2 |
| | | 100 | 0.836 | 0.607 | 0.83 | 8.08 | 0.551 | | |
| | RFdiffusion | 50 | 0.726 | 0.864 | 0.12 | 0.54 | 0.116 | **0.238** | 5 |
| | | 100 | 0.956 | 0.707 | 0.35 | 1.05 | 0.36 | | |
| Medium | Chroma | 200 | 0.76 | 0.676 | 0.48 | 4.68 | 0.275 | **0.296** | 3 |
| | | 300 | 0.684 | 0.637 | 0.67 | 5.73 | 0.398 | | |
| | | 400 | 0.545 | 0.668 | 0.74 | 6.92 | 0.262 | | |
| | | 500 | 0.48 | 0.685 | 0.84 | 8.56 | 0.249 | | |
| | FrameDiff | 200 | 0.694 | 0.666 | 0.39 | 1.82 | 0.524 | **0.399** | 2 |
| | | 300 | 0.682 | 0.656 | 0.48 | 2.38 | 0.486 | | |
| | | 400 | 0.5 | 0.654 | 0.55 | 3.27 | 0.378 | | |
| | | 500 | 0.448 | 0.69 | 0.59 | 4.7 | 0.209 | | |
| | RFdiffusion | 200 | 0.952 | 0.65 | 0.63 | 2.54 | 0.558 | **0.531** | 1 |
| | | 300 | 0.884 | 0.614 | 0.67 | 6.32 | 0.51 | | |
| | | 400 | 0.827 | 0.591 | 0.76 | 10.3 | 0.541 | | |
| | | 500 | 0.722 | 0.596 | 0.87 | 14 | 0.515 | | |
| Long | Chroma | 600 | 0.432 | 0.733 | 0.88 | 12.2 | 0.516 | **0.354** | 2 |
| | | 800 | 0.434 | 0.746 | 0.84 | 19.27 | 0.325 | | |
| | | 1000 | 0.427 | 0.742 | 0.7 | 24.67 | 0.222 | | |
| | RFdiffusion | 600 | 0.654 | 0.609 | 0.891 | 20.26 | 0.689 | **0.406** | 1 |
| | | 800 | 0.465 | 0.677 | 1 | 37.44 | 0.351 | | |
| | | 1000 | 0.384 | 0.738 | 1 | 54.43 | 0.178 | | |

## Motif-scaffolding

To benchmark the motif-scaffolding performance of different methods, we adopted the 24-cases benchmark set from RFdiffusion [34]. We provide the detailed descriptions and information of this benchmark set in S1 Table in S1 File including the motif regions and the corresponding PDB code for tested cases. Fig 5A displays the average designability of tested methods. We observed that RFdiffu*sion* demonstrated strong efficacy achieving an average success rate of approximately 40%, while *GPDL* and *TDS* generated considerably diverse backbones. All methods exhibited high variance in their distribution, implicating that their performances varied significantly across different tasks.

To explore the differences in success rate and diversity across distinct cases, we computed these metrics for each individual case (shown in S13-S14 Figs in S1 File). The results reveal that different methods have specific strengths in designing certain cases. Cases like *2KL8* (Fig 5B) could be designed effectively by all tested methods. Other cases, such as *7MRX_60* (Fig 5C), showed divergent performances across different methods. A few difficult cases like *4JHW* (Fig 5D) are challenging for most methods, presenting open problems for this task. The varying performance like results from

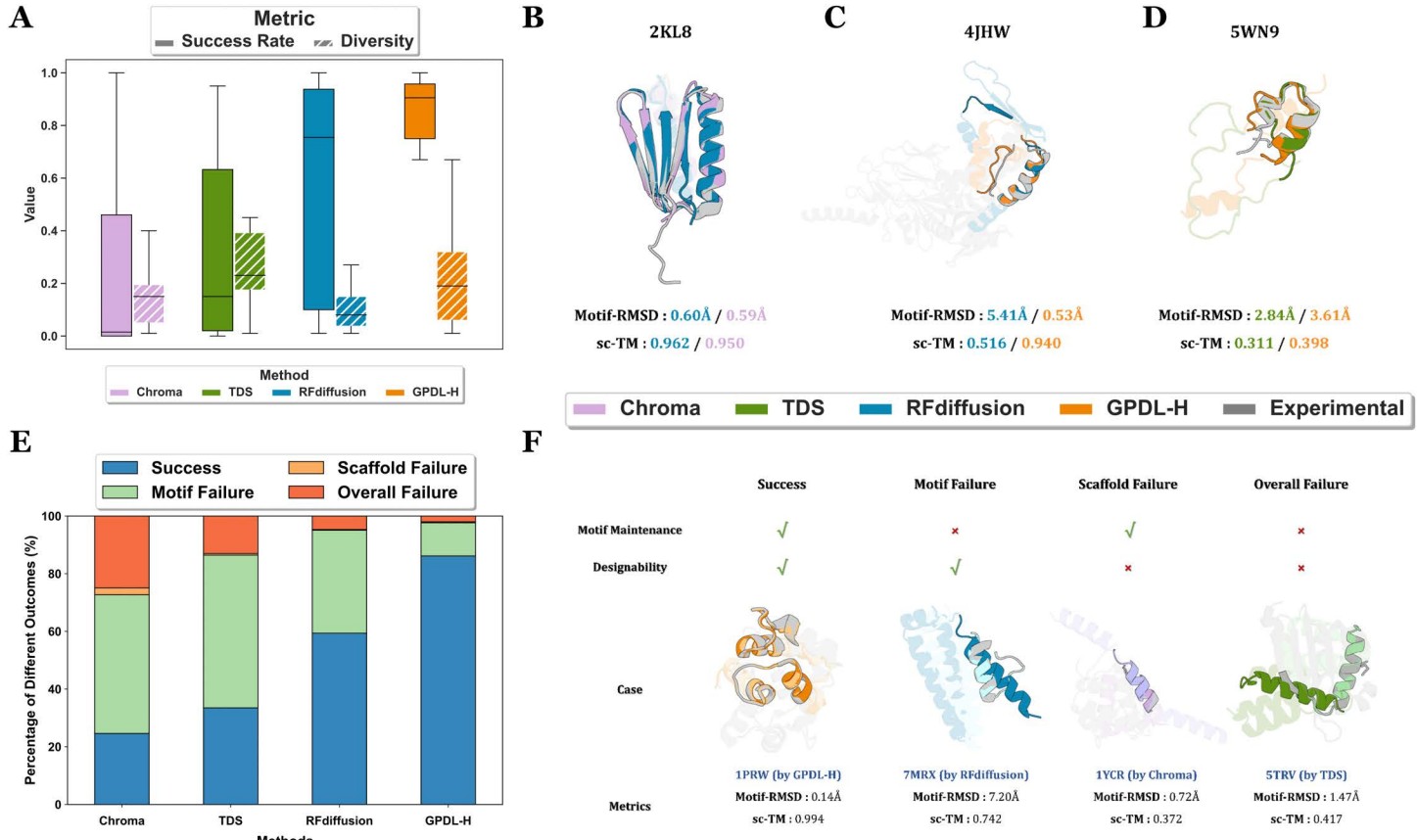

**Fig 5. Motif-scaffolding results and failure mode analysis for tested methods. (A)** Overall distribution of success rate and diversity across the 24-cases benchmark. The success definition follows the "Top N" strategy in unconditional generation. **(B)**, **(C)**, **(D)** The difficulty of solving motif-scaffolding problems varies across different cases. **(B)** An illustration of case *2KL8*. **(C)** An illustration of case *7MRX_60*. **(D)** An illustration case *4JHW*. In **(B)**, **(C)** and **(D)**, refolded structures of designs by different methods (shown in the legend on top of Fig 4F) are overlayed with crystal structures from PDB (shown in gray). Motifs are displayed as *opaque* and scaffolds are displayed as *transparent*. **(E)** Percentage of different outcomes of motif-scaffolding task among the 24-cases dataset. Each color denotes a unique type of outcome. **(F)** Schematic diagram of different outcomes. Each case in the "case" panel denotes a unique type of outcome and different colors denote different methods (shown in the legend on top of Fig 4F). Inside each scheme, we use *dark* colors to represent *refolded structures* after folding and *light* colors to denote *original backbones* generated by different methods. Experimental structures from PDB are shown in *gray*, motifs are displayed as *opaque* and scaffolds are displayed as *transparent*.

differences in the secondary structures and topologies of certain motifs, as well as the length scopes of motifs and scaffolds for different cases. For instance, the motif of *2KL8* has far more residues than its scaffold, allowing models to sculpt a small scaffold region more easily.

To gain a deeper understanding of the potential causes of unsuccessful designs beyond the explicit performances of different methods, we analyzed the outcomes of each motif-scaffolding task by grouping them into four categories: (1) *overall success* cases that satisfies both motif maintenance and scaffold designability; (2) *motif failure*, where the designs fulfill motif maintenance but not overall designability; (3) *scaffold failure*, where the designs meet overall designability but not motif maintenance, and finally (4) *overall failure*, where the designs fail to meet either criterion.

Results in **Fig 5E** suggest that *motif failure* dominated the failed cases followed by *overall failure*. Notably, *scaffold failure* accounted for a minimal fraction of failed cases, indicating that meeting the overall designability is necessary, but not sufficient, for a successful design.

The fact that overall designability is a prerequisite for solving the motif-scaffolding problems emphasizes the necessity for generating a reasonable protein structure compared to meeting a certain substructure. We propose that this concept can be generalized to the broader domain of conditional generation, where designing a physically plausible protein structure is more crucial than simply meeting certain conditions. Based on this idea, we hypothesize that extending a well-established unconditional generation model to conditional generation is more efficient than developing a conditional generation model from scratch, as the former approach benefits from a model that has already captured the underlying distribution of protein structures to some extent.

Motif-scaffolding has numerous applications in protein engineering, with one being the catalytic site conditioned motif-scaffolding which aims to design a novel enzyme scaffold starting with a set of catalytic motif residues. Therefore, we tested the motif-scaffolding methods with active sites of an enzyme, RA95.5-8F, which is an artificial retro-aldolase catalyzing a retro-aldol cleavage reaction. This enzyme was originally designed through directed evolution [81] and later redesigned using a well-established motif-scaffolding pipeline based on RFdiffusion, Riff-Diff [82], with a considerable success rate. Here we select this enzyme to see whether current motif-scaffolding methods are capable of scaffolding such enzyme catalytic sites starting with minimal inputs. Specifically, aside from the four tested methods described above, we additionally use an active site-specialized version RFdiffusion which was fine-tuned with minimalist sites along with implicit modeling of the substrate with an external potential, which we termed "RFdiffusion-enzyme". We intended to explore whether this strategy could shed light on the catalytic enzyme scaffolding task.

The success rates of five tested methods are displayed in S3 Table in S1 File. We found the base versions for all four methods failed to produce a single solution for the active site of retro-aldolase, highlighting the difficulty of scaffolding the motifs containing multiple discontinuous residues. We hypothesize this could be due to the fact that the information contained in the input catalytic residues is too less as the constraint for the task of conditional generation. Surprisingly, we found RFdiffusion-enzyme succeeded 20 out of 100, which is a considerable success rate for such a challenging task. This highlights the importance for the minimalist site fine-tuning and the external information from the substrate. We suggest that these could be insightful strategies for method developments of the conditional generation task in the future, particularly for methods tailored for minimalist sites motif-scaffolding, such as the enzyme design task.

## Discussion

In this study, we systematically evaluated protein backbone generation methods across diverse tasks, highlighting two major challenges: diversity in short proteins and designability in long proteins. While novelty has been overemphasized in prior studies, its practical relevance to biological applications appears limited, as methods showed minimal disparities in this regard. Although our evaluations were conducted *in silico*, the structured framework we developed provides valuable insights into the strengths and weaknesses of current methods. This framework can identify designable scaffolds, facilitating high-throughput filtering for downstream in vitro experiments.

Designing extremely short and large proteins is particularly useful in specific contexts, such as peptide drugs [83] or mini-binders [84,85] in the case of short proteins, and for large multi-domain proteins [86] or symmetrical protein assemblies [87,88] in the case of long proteins. While several peptide [89] and binder design [90] methods have been proposed to address these needs, designing large proteins remains an ongoing challenge. Furthermore, methods focusing on generating protein complexes or dynamic backbones critical for the design of enzymes of allosteric proteins are still in an early stage of development.

The fact that *motif failure* accounts for the majority of failures indicates that meeting certain conditions remains challenging for these models. While several sequence design methods have made progress in this area [48,91], backbone generation methods still require further refinement. We propose that clearer definitions of various conditional generation problems, along with customized training strategies like amortization, could significantly enhance the capacities of these methods. Despite the rise of powerful generated models, the considerable performance of *GPDL* demonstrates that

PLOS Computational Biology

hallucination-based methods [92–94] remain competitive. Given the high-throughput demand for backbone generation demand in downstream tasks, improving efficiency is another promising avenue for exploration.

We have also identified key areas for future exploration. Alternative modeling techniques for proteins such as sequence-structure co-design and all-atom generation, could address the limitations in current evaluation methods. Furthermore, the distinctions between scaffold parameterizations, like $C_\alpha$-based, frame-based and all-atom-based, could also play a crucial role in the discrepancy of performances.

Our analysis revealed a bias towards generating structured, helix-rich proteins, with limited capability to generate loop contents, which could typically be flexible regions in proteins and have potential to interact with other biomolecules. These regions are typically studied through molecular dynamics (MD) simulation [95–97] or conformational sampling [98], which are often excluded from most training sets of protein backbone generation methods. We suggest that employing a higher-level representation for protein structures, as well as augmenting training data, would substantially improve the generalizability of these methods. With the rapid advancement in this field, we are witnessing an explosion of sophisticated tasks in protein backbone generation, presenting both greater challenges and exciting opportunities in protein discovery.

## Supporting information

**S1 File.** S1 Fig.Compared studies on different sequence design methods. To make a comparison between ProteinMPNN and ESM-IF1, we randomly selected 20 backbones generated for each tested length (from 50 to 1000) and designed 10 sequences for each backbone, resulting in a subset containing 9 (number of unique lengths) × 20 = 180 unique backbones and 180 × 10 = 1800 designed sequences. We then feed these sequences into ESMFold and calculated the values of self-consistency TM-score (sc-TM). (A) Correlation between ProteinMPNN and ESM-IF1. Each point in the plot denotes a unique backbone, *x* and *y* axis denote the sc-TM values origined from ProteinMPNN and ESM-IF1, respectively. The degree of color denotes the number of residues of each sequence. The value displayed on the lower right denotes the Spearman correlation between results from different sequence design methods. (B) Distribution of sc-TM values from ProteinMPNN and ESM-IF1 are displayed in a boxenplot. It can be observed that sequences obtained from ProteinMPNN exist significant higher sc-TM values than ESM-IF1 no matter from the perspective of single backbones or overall distribution. S2 Fig. Correlation between three protein structure prediction methods. For comparison, we randomly selected sequences designed by ProteinMPNN from generated backbones and fed them into three structure predictors. For each length of unconditional sampled backbones, we randomly selected 100 sequences, which resulted in a total number of 900 sequences predicted by ESMFold, AlphaFold2 (without MSA) and OmegaFold, respectively. Each point in the subplot denotes a unique sequence and the value displayed on the lower right denotes the Spearman correlation between results from different structure predictors. The degree of color denotes the number of residues of each sequence. (A) Correlation between ESMFold and AlphaFold2 (without MSA). (B) Correlation between AlphaFold2 (no MSA) and OmegaFold. (C) Correlation between ESMFold and OmegaFold. S3 Fig. Designability distribution defined by sc-RMSD < 2.0 Å for different methods. As the same of Fig 2A, the violin plots display the distribution of sc-RMSD for different methods under each of the category. As the length arises, the performance on designable proportion downgraded sharply, with almost no designable backbones detected within the *long* group. S4 Fig. Success rate defined by sc-RMSD < 2.0 Å for different methods. As a comparison, the success rate here is defined by the number of backbones passing the sc-RMSD threshold (top 3/ best sc-RMSD < 2.0 Å) divided by the total number of backbones within each group, with top 3 sc-RMSD by solid line and best sc-RMSD by dashed line. The text of numbers refers to the solid lines. S5 Fig. Distribution of top 3/ best sc-RMSD for different methods. The white diamonds denote the median value of each group and the dashed line shows the designability threshold of sc-RMSD = 2.0 Å. (A) Distribution of top 3 sc-RMSD. (B) Distribution of best sc-RMSD. S6 Fig. Distribution of best sc-TM for different methods. The white diamonds denote the median value of each group and the dashed line shows the designability threshold of scTM = 0.5. S7 Fig. AUC for dynamic designability threshold, using sc-RMSD and

selecting the top 3 sequences for each backbone. The values on x-axis denotes the designability threshold of different sc-RMSDs, where the success definition is "top 3" sc-RMSD $<k$ Å when x = $k$. We take the step of sc-RMSD as 0.1 Å during the calculation and used *Scipy* [66] to calculate the area under the curve. The AUC values of different methods are shown inside the legend of each subplot. S8 Fig. AUC for dynamic designability threshold, using sc-TM and selecting the best sequence for each backbone. The values on x-axis denotes the designability threshold of different sc-TMs, where the success definition is "best" sc-TM> $k$ when x = $k$. We take the step of sc-TM as 0.01 during the calculation and used Scipy to calculate the area under the curve. The AUC values of different methods are shown inside the legend of each subplot. S9 Fig. AUC for dynamic designability threshold, using sc-RMSD and selecting the best sequence for each backbone. The values on x-axis denotes the designability threshold of different sc-RMSDs, where the success definition is "best" sc-RMSD $<k$ Å when x = $k$. We take the step of sc-RMSD as 0.1 Å during the calculation and used *Scipy* to calculate the area under the curve. The AUC values of different methods are shown inside the legend of each subplot.S10 Fig. Clash scores of backbones for different methods calculated by MolProbity. The vertical dashed lines denote the mean values for backbones within each method. S11 Fig. Percentage of bad covalent bonds of backbones for different methods calculated by MolProbity. The vertical dashed lines denote the mean values for backbones within each method. S12 Fig. Percentage of bad angles of backbones for different methods calculated by MolProbity. The vertical dashed lines denote the mean values for backbones within each method. S13 Fig. Detailed Motif-scaffolding performances for tested methods. Each facet places the result of one specific cases of four methods. (A) Success Rates of different methods on designability. The success threshold is defined as refolded sc-TM of the whole structure $>0.5$ and motif-RMSD$<1.0$ Å. The definition of success rate follows the ones with unconditional generation, with the areas with pure color denote the top 3 success rates and the shadow areas as the value surpassed by best success rates. (B) Diversity of different methods. The lollipops display the diversity values of sdifferent methods. S14 Fig. Motif-scaffolding performances using backbone success as sc-RMSD$<2.0$ Å. Each facet places the result of one specific cases of four methods. The success definition follows the ones in Fig 4 except changing the backbone success by sc-RMSD. The overall success threshold is defined as refolded sc-RMSD of the whole structure$<2.0$ Å and motif-RMSD$<1.0$ Å. The areas with pure color denote the top 3 success rates and the shadow areas as the value surpassed by best success rates. S15 Fig. Correlation between different metrics. The points displayed here are all protein backbones generated in the task of unconditional generation, where different colors denote the length of protein. The Spearman correlation value between two metrics is displayed on the plot. (A) Correlation between ESMFold pLDDT and self-consistency TM-score. (B) Correlation between ESMFold pAE and self-consistency TM-score. S16 Fig. Diversity results for unconditional generation using different TM-score threshold by Foldseek Cluster. (A) Diversity values of TM-score threshold$=0.4$. (B) Diversity values of TM-score threshold$=0.6$. S1 Table. Benchmark cases selected for evaluation on motif-scaffolding. Case: Different cases are denoted by their PDB IDs. We basically followed the benchmark set collected by *RFdiffusion* except *6VW1* since it is a complex generation task which *TDS* was not capable to perform. Input: The part prefixed by a letter in bold are the inputs (chain, residues) from the real PDB structure provided to the model (the "functional-site" or namely the "motif"). The part without a letter prefix are the lengths that the different methods randomly sampled to generate good designs (the non-functional site or namely the "scaffold"). Total Length: This indicates the total lengths of protein backbones output by different models. For *Chroma*, we performed the experiment with its SubstructureConditioner. For the other three method, we followed their original experimental settings except fixing the types of amino acids of all motif positions without redesign. The different experimental settings resulted in different sets of total lengths, where *RFdiffusion* and *Chroma* could perform length-variable design by randomly sampling from some specific length scopes, while *TDS* and *GPDL* generated proteins with mostly fixed lengths. Note that the differences between lengths of fixed or random lie on the scaffold parts, where all method were input with the same motifs from real protein structures. S2 Table. Significant test between whether adding the low-designability penalty during novelty calculation. A one-sided Mann-Whitney U Test was performed within each category along different lengths and methods. S3 Table. Performances of the motif-scaffolding methods to scaffold the catalytic site of retro-aldolase.

RFdiffusion-enzyme refers to the version using active site fine-tuned checkpoint with external substrate potential; 100 backbones was generated and evaluated for each method.
(DOCX)

## Acknowledgments

We would like to thank Brian L. Trippe for contributing to the codebase and helpful feedbacks on the manuscript, Xiaochen Cui, Xiaoyue Ji, Taeyoung Choi for insightful comments, and Tao Guo for helpful discussions on practical usage related to Foldseek-Cluster.

## Author contributions

**Conceptualization:** Hai-Feng Chen.

**Data curation:** Zhuoqi Zheng, Bo Zhang.

**Funding acquisition:** Hai-Feng Chen.

**Investigation:** Zhuoqi Zheng, Ting Wei.

**Methodology:** Zhuoqi Zheng, Bo Zhang, Bozitao Zhong, Jinyu Yu, Kexin Liu, Zhengxin Li, Junjue Zhu.

**Resources:** Zhuoqi Zheng.

**Supervision:** Ting Wei, Hai-Feng Chen.

**Validation:** Zhuoqi Zheng.

**Writing – original draft:** Zhuoqi Zheng.

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
