## [Decision Letter · Decision Letter 0]

1 Feb 2026

PCOMPBIOL-D-25-02477

Scaffold-Lab: Critical Evaluation and Ranking of Protein Backbone Generation Methods in A Unified Framework

PLOS Computational Biology

Dear Dr. Chen,

Thank you for submitting your manuscript to PLOS Computational Biology. After careful consideration, we feel that it has merit but does not fully meet PLOS Computational Biology's publication criteria as it currently stands. Therefore, we invite you to submit a revised version of the manuscript that addresses the points raised during the review process.

We look forward to receiving your revised manuscript.

Kind regards,

Gaetano Montelione, Ph.D.

Academic Editor

PLOS Computational Biology

Nir Ben-Tal

Section Editor

PLOS Computational Biology

**Journal Requirements:**

1) Please provide an Author Summary. This should appear in your manuscript between the Abstract (if applicable) and the Introduction, and should be 150-200 words long. The aim should be to make your findings accessible to a wide audience that includes both scientists and non-scientists. Sample summaries can be found on our website under Submission Guidelines:

Potential Copyright Issues:

i) Figure 1. Please confirm whether you drew the images / clip-art within the figure panels by hand. If you did not draw the images, please provide (a) a link to the source of the images or icons and their license / terms of use; or (b) written permission from the copyright holder to publish the images or icons under our CC BY 4.0 license. Alternatively, you may replace the images with open source alternatives. See these open source resources you may use to replace images / clip-art:

6) Please ensure that the funders and grant numbers match between the Financial Disclosure field and the Funding Information tab in your submission form. Note that the funders must be provided in the same order in both places as well.

**Reviewers' comments:**

Reviewer's Responses to Questions

**Comments to the Authors:**

Reviewer #1: On pg. 4 of SI it appears that some text might be missing. The authors write:

"We additionally adapted two modifications: "

after which the text moves onto a new paragraph.

Reviewer #2: In this manuscript, “Scaffold-Lab: Critical Evaluation and Ranking of Protein Backbone Generation Methods in A Unified Framework,” Authored by Zheng et al., the authors comprehensively evaluated protein backbone generation methods across various protein sizes and scaffolds and provided a systematic framework for evaluating Protein design methods. They have used extensive properties to rate and rank the available methods for protein design into multiple categories. They found that RFdiffusion performed most consistently across proteins of varying lengths. However, based on their evaluation metrics, the authors have concluded that none of the discussed methods is well-suited to all categories. Overall, these methods are biased towards generating more structured proteins and have limited capabilities to design loops and intrinsically disordered regions. They suggest users must design a framework tailored to the length and biological function of their protein sequence.

An interesting point to note is that, according to the evaluation in this paper, the authors conclude that novelty in a designed protein structure comes at the expense of its designability, questioning the emphasis on novelty in protein backbone generation methods. In general, low designability indicates that the structure is farther from the existing protein, and hence, the feasibility of in vivo expression of such proteins might be problematic. Hence, the authors have questioned the overemphasis on the novelty of designed proteins and suggest that designability of the motif scaffold is a prerequisite when users are interested in a certain biological function.

Reviewer #3 (Academic Editor)

Zheng et al describe Scaffold-Lab, a unified framework for systematic evaluation of protein  backbone generation methods. They evaluated several representative methods and provide a detailed analysis of their performance and utilities. Overall, RFdiffusion performed the best among all tested methods. Other methods also demonstrated reliable performance on short (< 300 residues) proteins

Considering the rapid advances in AI methods for de novo protein design, the authors are correct to point out there is a  need for standardized benchmarks to assess protein backbone generation methods. Scaffold-Lab provides unified framework for the evaluation of protein backbone generation methods. The results clearly demonstrate the challenge for most methods is "designability" for targets > 300 residues, and the unique strengths and weaknesses of RFDiffusion. The criteria for assessing various methods and protocols are also useful. Although the study is not as comprehensive as one might like, it does provide a significant and valuable effort in this direction.

A revised manuscript should address the following issues.

1. The authors state in the final paragraph of the Introduction: "We observed that the ability to create an ideal scaffold is a prerequisite of successful designs than simply meeting certain conditional criteria" The language here is not clear. Since this is a key point - it should be made more clearly.

2. For "structural properties", since the proteins are refolded with ESM_Fold or AlphaFold, could the authors also consider an empirical packing score like Molprobity? This has the advantage also that it is quantifiable.

3. In Methods the author state "For each unique length within each method, we generated 100 backbones with 10 sequences for each backbone to be evaluated through a refolding pipeline." This seems to relate to the Results of Fig 2 and 3, but not for the motif-scaffolding study which uses the 24-case benchmark of RFdiffusion. Perhaps this point is better made in the legends of Fig 2 and 3

4. Panel 2F. The only stars shown are ***. It appears contradictory that the distributions for Medium RFDiffusion with and without penalty look nearly the same, but have p < 0.001, while for Medium FrameDiff the distributions look different but are not statistically different. Is this all correct?

5. Both the Analysis of Structural Properties and the last paragraph of Discussion do not properly distinguish "ordered non-regular structures" from "intrinsically-disordered" regions. These are treated as synonymous, which they are not. Are the designed regions designated as "coil" non-regular ordered regions or disorder regions?

6. Ranking. It would be nice to have a histogram plot for Mean Ci vs method, ordered by decreasing Ci score. This would make it clear which methods were best scoring. One histogram plots could combine the data for short, medium, and long lengths.

7. The observed correlation between successful non-conditional designability and successful conditional motif scaffolding is very important, though perhaps obvious to some. The paper would be stronger if more additional assessments were made of different approaches to conditional design, and/or applications to specific cases of binder or active site design.  It seems that exploring strengths and weaknesses of different methods and their parameter choices in conditional design is a tremendous value of Scaffold-Lab.  Although additional studies addressing this are not required, the authors should at least discuss this point in a bit more detail.

8. Although a web server is not provided, there appears to be an extensive github site which will allow scientists to use Scaffold-Lab in their research

Minor points

page 3. Cartesian should be capitalized

Supplementary Materials should be capitalized throughout

Fig.3  Define "coll percentage" on Y axis

At several points, the English needs some refinement. For example:

page 17. "Recently studies leveraging diffusion models to generate intrinsically disordered regions of proteins have shown great potentials applicated in both protein..."

Fig 3 legend. "All data was curated by directly calculating onto protein backbones without refolding"

**Have the authors made all data and (if applicable) computational code underlying the findings in their manuscript fully available?**

Reviewer #1: Yes

Reviewer #2: Yes

Reviewer #3: Yes

PLOS authors have the option to publish the peer review history of their article (what does this mean?). If published, this will include your full peer review and any attached files.

Reviewer #1: No

Reviewer #2: No

**Figure resubmission:**
---

## [Decision Letter · Decision Letter 1]

4 May 2026

Dear Prof. Dr. Chen,

We are pleased to inform you that your manuscript 'Scaffold-Lab: Critical Evaluation and Ranking of Protein Backbone Generation Methods in A Unified Framework' has been provisionally accepted for publication in PLOS Computational Biology.

Best regards,

Gaetano Montelione, Ph.D.

Academic Editor

PLOS Computational Biology

Nir Ben-Tal

Section Editor

PLOS Computational Biology

Reviewer's Responses to Questions

**Comments to the Authors:**

Reviewer #1: The MolProbity clash score analysis is a welcome addition to the paper and clearly highlights the advantages imparted by design models that contain van der Waal loss penalizing functions.

The the active site-conditioned design benchmark nicely highlights some of the present difficulty in designing enzymes using this approach.

The addition of a Colab Notebook environment will also greatly enhance accessibility and dissemination of this work.

Reviewer #3: The authors have adequately addressed all the issues I raised.

**Have the authors made all data and (if applicable) computational code underlying the findings in their manuscript fully available?**

Reviewer #1: None

Reviewer #3: Yes

PLOS authors have the option to publish the peer review history of their article (what does this mean?). If published, this will include your full peer review and any attached files.

Reviewer #1: No

Reviewer #3: **Yes:** Gaetano T. Montelione

---

## [Editor Report · Acceptance letter]

PCOMPBIOL-D-25-02477R1

Scaffold-Lab: Critical Evaluation and Ranking of Protein Backbone Generation Methods in A Unified Framework

Dear Dr Chen,

I am pleased to inform you that your manuscript has been formally accepted for publication in PLOS Computational Biology. Your manuscript is now with our production department and you will be notified of the publication date in due course.

With kind regards,

Aiswarya Satheesan
